# Sequencing B cell receptors from ferrets (*Mustela putorius furo*)

**Julius Wong**[1], **Celeste M. Tai**[2], **Aeron C. Hurt**[2], **Hyon-Xhi Tan**[1], **Stephen J. Kent**[1,3,4]*,
**Adam K. Wheatley**[1,4]*

**1** Department of Microbiology and Immunology, University of Melbourne, at The Peter Doherty Institute for Infection and Immunity, Melbourne, Victoria, Australia, **2** World Health Organization (WHO) Collaborating Centre for Reference and Research on Influenza, The Peter Doherty Institute for Infection and Immunity, Melbourne, Victoria, Australia, **3** Melbourne Sexual Health Centre and Department of Infectious Diseases, Alfred Hospital and Central Clinical School, Monash University, Melbourne, Victoria, Australia, **4** ARC Centre for Excellence in Convergent Bio-Nano Science and Technology, University of Melbourne, Parkville, Victoria, Australia

* a.wheatley@unimelb.edu.au (AKW); skent@unimelb.edu.au (SJK)

**Data Availability Statement:** All relevant data are within the manuscript and its Supporting Information files.

**Funding:** The Melbourne WHO Collaborating Centre for Reference and Research on Influenza is

## Abstract

The domestic ferret *(Mustela putorius furo)* provides a critical animal model to study human respiratory diseases. However immunological insights are restricted due to a lack of ferret-specific reagents and limited genetic information about ferret B and T cell receptors. Here, variable, diversity and joining genes within the ferret kappa, lambda and heavy chain immunoglobulin loci were annotated using available genomic information. A multiplex PCR approach was derived that facilitated the recovery of paired heavy and light chain immunoglobulin sequences from single sorted ferret B cells, allowing validation of predicted germline gene sequences and the identification of putative novel germlines. Eukaryotic expression vectors were developed that enabled the generation of recombinant ferret monoclonal antibodies. This work advances the ferret as an informative immunological model for viral diseases by allowing the in-depth interrogation of antibody-based immunity.

## Introduction

Effective humoral immunity is contingent upon the phenomenal diversity of antibodies. In mammals, this is derived via genetic recombination of numerous variable (V), diversity (D) and joining (J) gene segments localised to heavy, kappa and lambda immunoglobulin loci. In recent years, the capacity to clone and express antibodies from single B cells has proved a powerful tool to study antibody repertoires in a variety of infectious disease settings in humans [1–4], and important animal models such as mice [5, 6] and non-human primates [7, 8]. These approaches have subsequently been extended using next-generation sequencing platforms (reviewed in [9, 10]), allowing unprecedented depth in the characterisation of anti-pathogen antibody responses.

The domestic ferret *(Mustela putorius furo)* is a critical mammalian model to study pathogenesis and evaluate vaccines against a variety of human respiratory pathogens (reviewed in [11]), most critically influenza. However, the majority of influenza research using ferrets is

supported by the Australian Government Department of Health. This work was supported by NHMRC programme grant #1052979 (SJK) and NHMRC project grant #1129099 (AKW). JW is supported by a Melbourne International Research Scholarship and Melbourne International Fee Remission Scholarship.

**Competing interests:** The authors have declared that no competing interests exist.

focused upon viral transmission and/or pathogenesis, with in-depth immunological studies limited by a limited understanding of the ferret immune system (reviewed in [12]). A key knowledge gap surrounds the immunogenetics of ferret immunoglobulins. While the ferret genome was recently sequenced [13] accurate annotation of germline immunoglobulin genes is currently incomplete. This has hindered the ability to sequence ferret B cell receptors and/or allow the recovery of ferret monoclonal antibodies, limiting detailed interrogation of ferret serological responses that informs current influenza vaccine strain selection efforts.

Here we sought to increase the utility of ferrets for studying humoral immunity. Ferret heavy, kappa and lambda immunoglobulin loci were annotated using available genomic sequences, allowing the design of a novel set of multiplex PCR primers flanking recombined ferret immunoglobulin genes. Recombined B cell receptor sequences were recovered from single sorted ferret B cells, partially confirming our initial gene segment annotation and allowing identification of potential novel germlines. Ferret immunoglobulin constant gene sequences were confirmed using *de-novo* assembly of RNA-seq transcripts, allowing the design of expression plasmids and the recombinant production of ferret IgG monoclonal antibodies. In summary, we present a single-cell, RT-PCR based approach for recovery of B cell receptor immunoglobulins from ferret B cells and the recombinant production of ferret monoclonal antibodies in vitro, analogous to methodologies in widespread use in rodents and primates.

## Materials and methods

### Annotation of ferret immunoglobulin loci

Ferret genomic contigs containing potential immunoglobulin genes were retrieved from *e! Ensembl* (http://www.ensembl.org). (Immunoglobulin heavy loci—GL897360.1, GL897427.1, GL897453.1, GL897498.1, GL897556.1, GL897558.1, GL897564.1, GL897795.1, GL898421.1; kappa loci—GL896905.1; lambda loci—GL897406.1, AEYP01111698.1, GL896906.1, AEYP011112098.1, GL897285.1, GL897406.1, GL897344.1, GL897565.1, GL897418.1, AEYP01110728.1, AEYP01108526.1, GL897638.1 GL897285.1, GL897484.1, GL897019.1, GL897400.1). Iterative BLAST searches using human, and then ferret immunoglobulin gene segments were used to identify and annotate putative germline genes. Ferret immunoglobulin gene sequences were analysed with reference to human, mouse or canine databases using IMGT/V-Quest [14] and assigned to mammalian clans based upon phylogenetic analyses. Sequences with nonsense mutations and/or non-functional regulatory elements were considered pseudogenes. Phylogenetic relationships of functional V genes were determined based on the Jukes-Cantor model. Consensus phylogenetic trees were built using the Neighbour-Joining method with no outgroups and resampled by bootstrapping using Geneious tree builder (10.1.3). Ferret V, D, J and constant gene sequences have been uploaded to Genbank.

### Flow cytometric sorting of single ferret B-lymphocytes

Ferret studies and related experimental procedures were approved and conducted in accordance to the University of Melbourne Animal Care and Use Standards by the relevant ethics committee (#CT-FER-17-05). Single cell suspensions were prepared from the spleen of immunologically naïve ferrets. PBMCs were purified using 95% Ficoll-Paque Plus and cryopreserved in heat-inactivated fetal calf serum (FCS) containing 10% dimethylsulfoxide (DMSO). Cryopreserved ferret PBMCs were thawed, stained with Live/Dead Fixable Aqua (Thermo Fisher), surface stains anti-CD11b-BV510 (Biolegend: clone M1/70), anti-CD8-BV450 (Thermo Fisher: clone OKT8) and anti-ferret IgA/IgM/IgG-FITC (Rockland Immunochemicals cat.618-102-130). Stained cells were resuspended in OptiMEM (Thermo Fisher) before single,

live, surface Immunoglobulin positive B cells were sorted into 96-well PCR plates and stored at -20˚C.

For the recovery of antigen-specific ferret B cells, a single ferret was infected with 1000 TCID$_{50}$ of H1N1 A/California/04/2009 and a single cell suspension of parapharyngeal lymph node cells (pLN) was prepared at 28 days post-infection and cryopreserved in heat-inactivated FCS containing 10% DMSO. Cells were subsequently thawed and stained with Live/Dead Fixable Aqua (Thermo Fisher), surface stains anti-CD11b-BV510 (Biolegend: clone M1/70), anti-ferret IgA/IgM/IgG-FITC (Rockland Immunochemicals cat.618-102-130), anti-CD8 eFluor450 (eBioscience Clone OKT8) and a prototype IgD Mab conjugated to APC-Cy7,. Biotinylated recombinant full length A/California/04/2009 hemagglutinin (HA) probes [15] conjugated to streptavidin-PE or streptavidin-APC (Invitrogen) were used to sort single HA-specific B cells into 96-well plates and stored at -20˚C.

## Ferret B cell receptor (BCR) sequencing

A B cell receptor (BCR) sequencing protocol was developed based upon the multiplex nested RT-PCR approaches previous described for humans [1] and mice [5]. Reverse transcription of total cellular RNA from single sorted ferret B cells was performed in 25μL reaction volumes in the sort plate using 450 ng random hexamers (Bioline), 50 U Superscript III (Thermo Fisher), 1X First Strand buffer (Thermo Fisher), 8 U RNAsin (Promega), 0.125pmol Dithiothreitol (DTT) (Thermo Fisher), 0.8% v/v IGEPAL CA-630 (Sigma Aldrich) and 0.8mM deoxynucleotide triphosphate (dNTP) (Bioline). Cycling conditions for cDNA synthesis were: 42˚C for 10 min, 25˚C for 10mins, 50˚C for 60 mins and 94˚C for 5min. 3μL of cDNA was used as template in multiplex nested PCR reactions to amplify paired recombined ferret heavy and light chain (IgK, IgL) sequences. Primary reactions were carried out in 50μL volumes using Hotstart Taq plus polymerase (Qiagen), 1X reaction buffer, 2.0 mM MgCl$_2$, 0.1mM dNTP (Bioline) and 5 nmol each of primary forward and reverse primer pools (S1 Table). 4μL of primary PCR product was used as template in a secondary, nested PCR (50 μL volume) containing 1 X reaction buffer, 1.5mM MgCl$_2$, 1 x solution Q and 5 nmol secondary forward and reverse primer pools (S1 Table). For recovery of antigen-specific B cells from an infected ferret, the protocol was amended by substituting primary and secondary heavy chain reverse primer pools with 5 nmol reverse primer binding in the heavy chain joining gene (IGHJ) (S1 Table). Cycling conditions for both heavy and light chain amplification are as follows: Primary PCR: 95˚C for 5 mins, 50 cycles of 95˚C for 30s, 55˚C for 30s and 72˚C for 55s. Final extension for 7 mins at 72˚C. Secondary PCR: 95˚C for 5 mins, 50 cycles of 95˚C for 30s, 50˚C for 30s and 72˚C for 60s. Final extension for 7 mins at 72˚C. PCR products were visualised by agarose gel electrophoresis prior to standard purification and standard sanger sequencing using the reverse constant chain primer (IgM, IgK, IgL) from the secondary amplification steps or the IGHJ reverse primer.

## Immunoglobulin gene sequence analysis

Full length $V_H$, $V_\kappa$ and $V_\lambda$ sequences recovered from single sorted ferret B-cells were analysed using IMGT/High V-Quest. As well-validated germline ferret immunoglobulin sequences are not yet available, sequences were compared against dog (*Canis lupus familiaris*) germline immunoglobulin sequences to identify V(D)J segments with the highest identity. CDR-H3 sequences were determined by counting amino acid residues immediately after framework region 3 starting from the conserved cysteine (C) residue and ending with a conserved tryptophan (W) or phenylalanine (F).

## De-novo transcriptome assembly by RNA-Seq

RNA was extracted from five million cryopreserved PBMCs derived from a single ferret spleen using RNeasy Plus micro Kit (Qiagen). mRNA libraries were prepared using Illumina Truseq Stranded mRNA kit and 100bp single end reads were obtained using an Illumina HiSeq 3000. Analysis was performed using Galaxy (https://usegalaxy.org) [16]. Sequences were filtered (Q>30) and trimmed (Trim) to remove illumina adapter sequences. Contigs were assembled *de-novo* with > 40 bp minimum read overlap for path extension using Trinity [17]. Contigs were aligned to ferret genomic sequences or alternatively, canine immunoglobulin constant region homologues, using MAGIC-BLAST [18] to identify putative constant genes. Ferret constant region gene sequences were then aligned to human and canine homologues using Geneious 10.1.3.

## Cloning and expression of chimeric human/ferret and fully ferret monoclonal antibodies

Validated ferret IgG, and kappa and lambda constant domain sequences were synthesized (Geneart) and cloned into established human IgG1 and light chain eukaryotic expression vectors [1] using SalI/BamHI (IgG1), XhoI/BamHI (Lambda) or BsiWI/BamHI (Kappa) restriction digests, agarose gel purification and T4 DNA ligation (NEB:New England Biolabs).

Recombined heavy chain (VDJ) and lambda chain (VJ) sequences of the human anti-HA antibody CR9114 [19] or of recovered HA-specific ferret antibodies were synthesized (Geneart) and cloned into ferret IgG1 and lambda chain expression vectors using flanking AgeI/SalI (IgG) and AgeI/XhoI (lambda) restriction enzymes (NEB). Plasmid DNA was prepared using standard MIDI or MEGA kits (Qiagen).

For recombinant expression, heavy and lamdba chain expression plasmids were co-transfected into Expi293F cells using Expifectamine (Thermo Fisher). Briefly Expi293F cells were grown to a density of 2.5 x 10$^6$ cells/mL and transfected with 100 μg each of heavy and light chain expression plasmids complexed with 270 μL of Expifectamine. After addition of transfection enhancers 16 hours post-transfection, culture supernatants were harvested 5 days post-transfection, clarified by high speed centrifugation (5000 g, 4˚C, 15 min) and filtration (0.22μm). Antibodies were purified using Pierce Protein A agarose (Thermo Fisher). Briefly, supernatants were equilibrated with Protein A IgG binding buffer and loaded into purification columns packed with 0.2mL of equilibrated Protein A agarose. Bound antibody was washed using Protein A IgG binding buffer (Thermo Fisher), and eluted with 15mL IgG elution buffer (Thermo Fisher) and neutralised with 1.5mL 1M TE buffer pH8.0 (Merck). Antibodies were concentrated via centrifugation (100 kDa Amicon; Merck Millipore at 3000g, 4˚C, 15 min), resuspended in PBS and analysed using SDS-PAGE. Samples were denatured in 1x SDS loading buffer containing 5% v/v β-mercaptoethanol at 95˚C for 15 min, resolved using 16.5% precast polyacrylamide gel (Biorad) in 1 x SDS buffer (Biorad), fixed (40% Ethanol; 10% acetic acid) and stained with colloidal Coomassie blue (Biorad).

## Influenza hemagglutinin (HA) ELISA

96-well ELISA plates were coated with 400ng per well of full length recombinant CA09 HA at 4˚C overnight. Wells were then blocked with 2.5% BSA/PBS for 1h at room temperature and washed with PBS-Tween (0.05%) five times. Four-fold serial dilutions of recombinant monoclonal antibodies (starting at 100 μg/mL), or ferret serum samples (starting at 1 in 100 dilution) added at room temperature for 1h. Detection was performed by sequential staining with donkey anti-ferret IgG (Rockland cat. 618-101-012) and goat anti-donkey-HRP (Abcam cat.

Ab6988) at 1:2000 and 1:1000 dilutions respectively for 30 min each. $OD_{450}$ readings were obtained after addition of Sureblue TMB peroxidase substrate (Seracare) and TMB BlueSTOP Solution (Seracare).

### Viral escape asssay

The generation of viral escape mutants was based upon previously described protocols [20]. Briefly, 24-well plates were seeded with 2.5 x $10^5$ Madin Darby Canine Kidney (MDCK) cells per well to form confluent monolayers. The next day, serial dilutions of recombinant ferret mAbs were incubated with A/California/04/2009 virus for one hour at 37˚C in Flu-media (Dulbecco's Modified Eagle's Medium (DMEM) with 0.8% bovine serum albumin (BSA), 1% penicillin/streptomycin and 0.1% L-1-Tosylamide-2-phenylethyl chloromethyl ketone (TPCK)-treated trypsin), before adding the virus-antibody mixture to MDCK cells. After 2 to 3 days culture media supernatants from wells with visible cytopathic effect were collected and used to infect a fresh monolayer of MDCK cells in the presence of increasing concentrations of mAb. After serial passaging, culture supernatants were harvested, viral RNA was extracted and the HA gene sequenced. Putative mutant viruses were identified based upon sequence comparison to similarly passaged media-only controls.

## Results

### Annotation of ferret germline variable, diversity and joining genes

The initial publication of the ferret genome was reported in 2014 [13]. However, the assembly and annotation of immunoglobulin loci is currently incomplete. We therefore identified and annotated genomic contigs containing potential heavy chain variable (IGHV) genes using iterative rounds of BLAST. A total of 27 unique IGHV-like genes were initially identified, including potential pseudogenes and/or non-functional segments. 20 IGHV genes retaining an open reading frame (ORF), downstream recombination signal sequence (RSS; heptamer and nonomer) and critical amino acid residues (eg. Cys 74) were considered potentially functional, although 4 of these lacked identifiable functional leader exons. Based on DNA sequence homology, three groupings of immunoglobulin genes, corresponding to the three vertebrate "clans" [21] could be delineated (Fig 1A): clan I (3 genes), clan II (1 gene) and clan III (16 genes). In line with reports from other *Carnivora* such as dogs [22] and cats [23], the majority of IGHV gene diversity in ferrets lies within Clan III (human IGHV3-21 or 3–69 homology), with sequence variance concentrated within the CDR-H1 and CDR-H2 regions (Fig 1B). Based upon the conserved arrangement of RSS sequences, we identified 7 putative D gene segments (numbered IGHD1-7) (Fig 1C), of which 3 appear orthologous to canine, murine and/ or human variants. Similarly, 5 putative germline J gene segments (IGHJ1-5) were identified (Fig 1D) including 2 conserved orthologues.

The 0.7Mb kappa chain locus is contained within a single contig (contig GL896905.1) allowing identification of 48 potentially functional germline IGKV genes (Fig 2A) using similar criteria as IGHV genes, in good agreement to previous computational approaches [24]. Based upon sequence homology with human and mouse genes, these could be divided among three clans; II (44 genes), I (2 genes) and III (2 genes) (Fig 2B).

Partial contigs bridging the lambda chain locus were extracted and analysed for lambda variable gene segments (Fig 2C). 32 potentially functional IGLV gene segments were identified, spanning four probable clans; I (2 genes), II (22 genes), IV (2 genes) and V (6 genes). Five potential kappa joining gene segments (IGKJ) and four lambda joining gene segments (IGLJ) were identified based on sequence homology to canine variants. Functionality was inferred based upon the presence of canonical junction F/W-G-X-G motifs, RSS elements and 5' donor

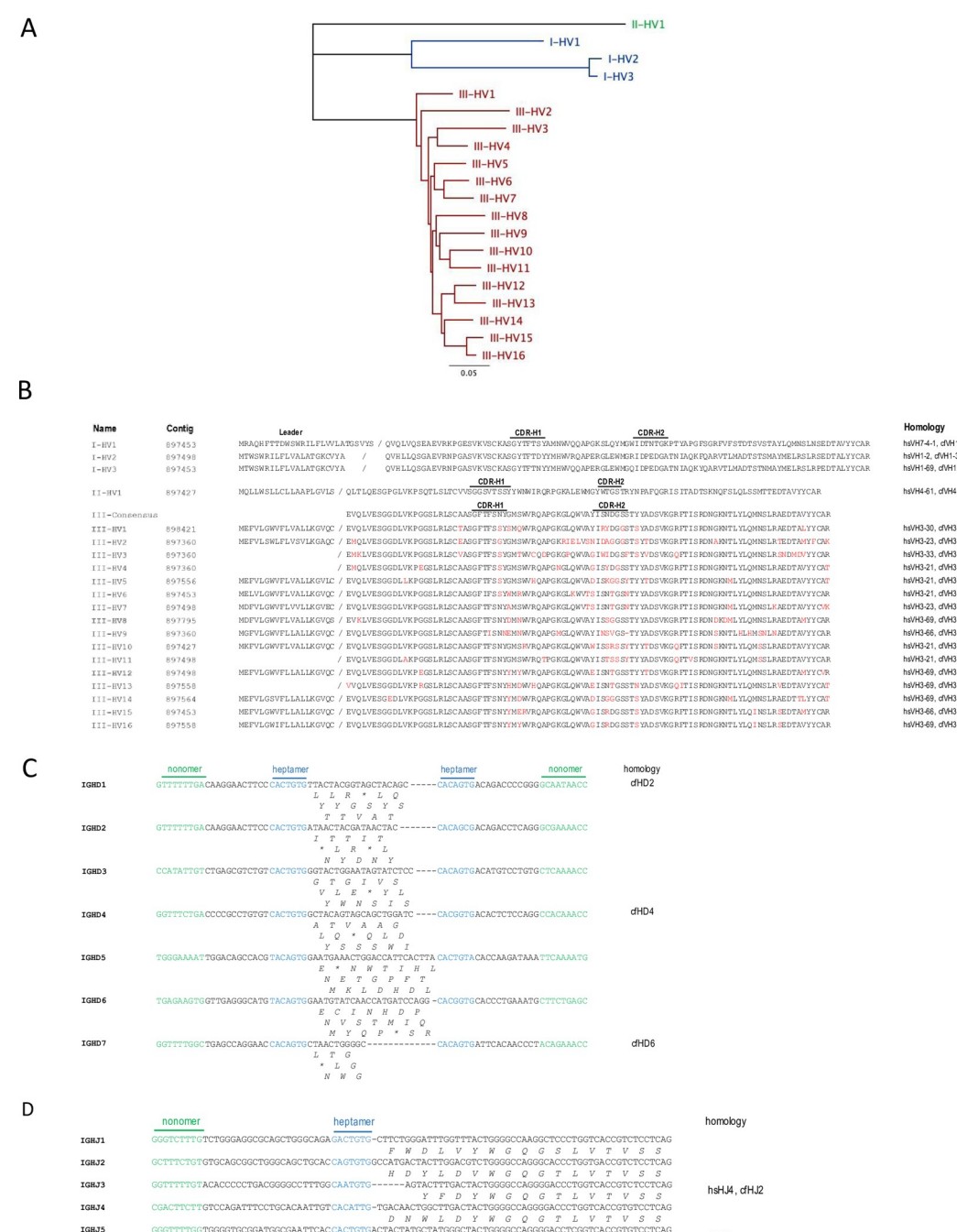

**Fig 1. Analysis of ferret immunoglobulin heavy chain gene segments.** (A) Non-rooted phylogenetic tree of putative ferret heavy chain variable gene segments. Branch lengths are proportional to genetic distance as indicated. (B) Leader and coding amino acid sequences of putative heavy variable gene (IGHV) segments. For clan III genes, amino acid residues highlighted in red are variable compared to the consensus. (C) Coding and RSS sequences of ferret heavy chain diversity gene (IGHD) segments. (D) Coding and RSS sequences of ferret heavy junction gene (IGHJ) segments.

splice sites (GTRAGT) in the adjacent intronic sequences. All putative heavy, kappa and lambda gene segments are provided in S1 Table.

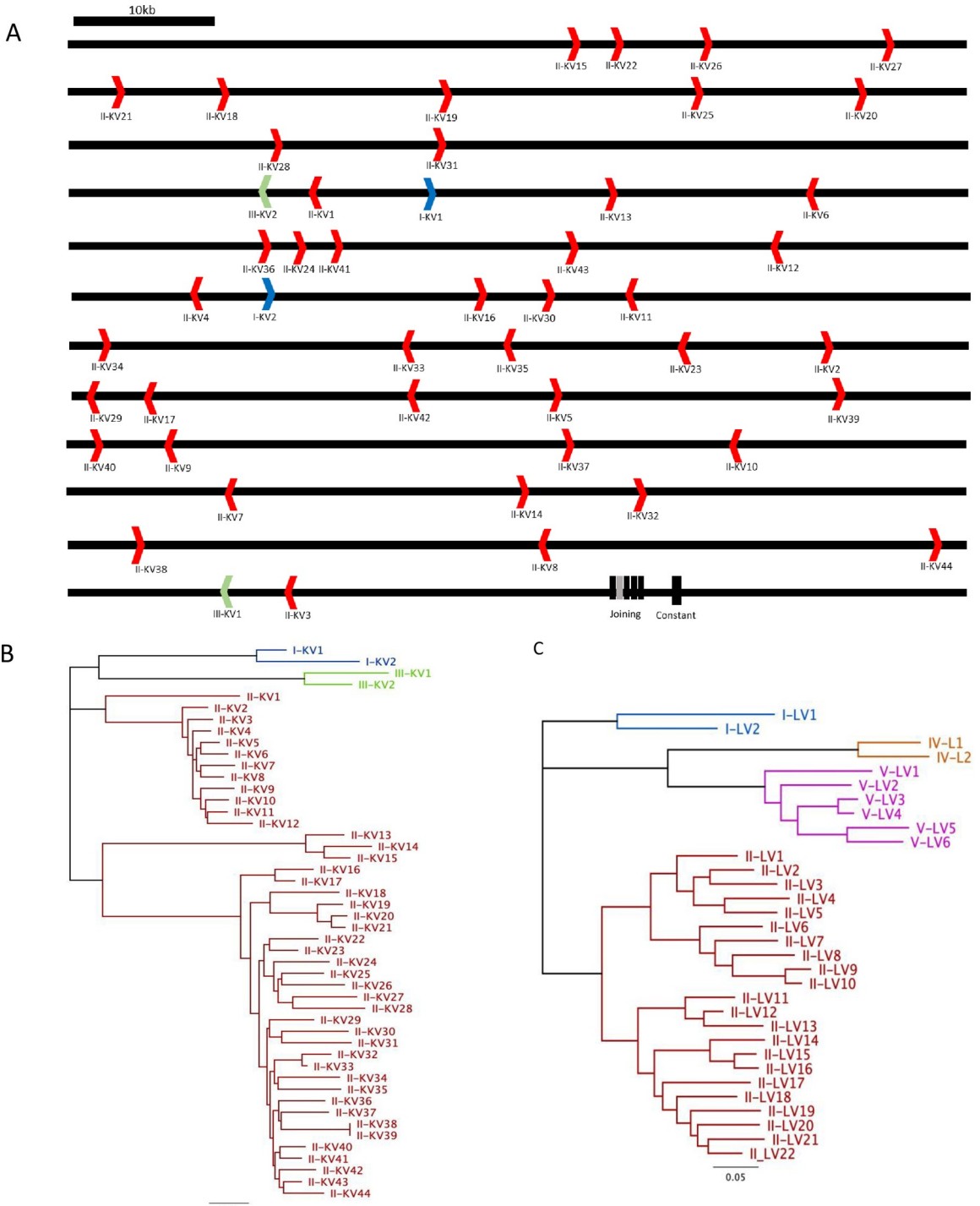

**Fig 2. Annotation of the ferret immunoglobulin light chain gene segments.** (A) Schematic of the kappa locus (contig GL896905.1) detailing the orientation of putative functional IGKV segments relative to the joining and constant genes (B) Non-rooted phylogenetic tree of kappa variable gene segments detailing the three potential clans identified. (C) Non-rooted phylogenetic tree of lambda variable gene segments detailing the four potential clans identified. Contig numbers are indicated in parentheses. Branch lengths are proportional to genetic distance as indicated.

## Single cell RT-PCR for recovery of ferret immunoglobulin heavy chain sequences

To investigate the expressed repertoire of germline immunoglobulin gene segments, we used the above sequence information to design multiplex PCR primers (S2 Table) targeting conserved leader and constant regions of recombined heavy, kappa and lambda mRNA transcripts. Using an approach analogous to other mammals [1, 5], single ferret B cells were sorted from cryopreserved splenocyte preparations by flow cytometry using a simple antibody panel and gating scheme (Fig 3A). cDNA was generated from each cell and immunoglobulin sequences recovered by nested multiplex PCR. A total of 121 functional recombined heavy chain sequences (all using a mu constant chain) were recovered from 480 sorted ferret B cells from three genetically outbred ferrets (~25% recovery). In line with the frequency of germline gene segments, the majority of immunoglobulins were derived from III clan genes, with only two sequences recovered from clan II and a single example of a sequence from clan I (Fig 3B).

An accurate assessment of germline IGHV gene utilisation is difficult due to (a) the current inability to segregate naïve versus B cells that are somatically mutated (memory) during the sort and (b) the limited genomic information surrounding the ferret IGHV locus and (c) a poor understanding of any allotypic variation within these outbred animals. Nevertheless, we recovered clusters of IGHV gene sequences that were highly conserved (>99% homology) to predicted germlines including examples from two of the three clans (II-HV1, III-HV7, III-HV8, III-HV9, III-HV12, III-HV16). A single clan I sequence was recovered, albeit with more limited homology to the putative germline gene (97.6%). The 18–20 IGHV genes we found within genomic contigs is less than observed in felines (24 genes) and canines (38 genes) [24], suggesting additional variable germlines may remain undiscovered. Supporting this, we repeatedly recovered multiple heavy chain immunoglobulin sequences that shared identical IGHV gene sequences, but recombined with different D and J genes, strongly suggestive of a common germline progenitor (III-HV17, III-HV18, III-HV19 and III-HV20, III-HV21). The existence and identity of these additional putative germlines (listed in S1 Table) will be clarified as genomic sequencing of ferrets continues. Utilisation of all five predicted IGHJ gene segments and 6 of 7 IGHD gene segments (not IGHD6) was evident within recombined BCR sequences. CDR-H3 regions, often a critical determinant for antigen recognition, ranged from 5 to 25 amino acids (mean 13.2) in length (Fig 3C); broadly comparable to canines [25] and potentially shorter on average than observed in humans [26].

## Recovery of ferret immunoglobulin light chain sequences

For ferret immunoglobulin light chains, 99/480 (20.6%) functional and recombined kappa sequences were recovered, all of which were derived from variable gene segments belonging to clan II and recombined with 4 of 5 predicted IGKJ gene segments (not IGKJ2) (Fig 4A). Germlines II-KV13, II-KV18, II-KV19 II-KV20, II-KV21, II-KV22. II-KV23, II-KV29, II-KV36, II-KV40, II-KV41, II-KV42 and II-KV43 were matched to the sequences (>99% identity) found *in silico*. Four additional novel germlines were evident within recovered sequences (II-KV45-48) (S1 Table). Given the kappa locus is fully annotated, these additional germlines most likely represent allelic variants of clan II germlines that differ between the outbred ferrets used for genomic sequencing versus B cell sorting.

For lambda chain immunoglobulins, 144/384 (37.5%) productive and recombined sequences were recovered, most homology to annotated clan II and clan I sequences (II-LVI1, II-LV2, II-LV4, II-LV5, II-LV6, II-LV7, II-LV8, II-LV11, II-LV12, II-LV14, II-LV15, II-LV16, II-LV17, II-LV18, II-LV20, I-LV1; Fig 4A). Additional potential germline genes were also

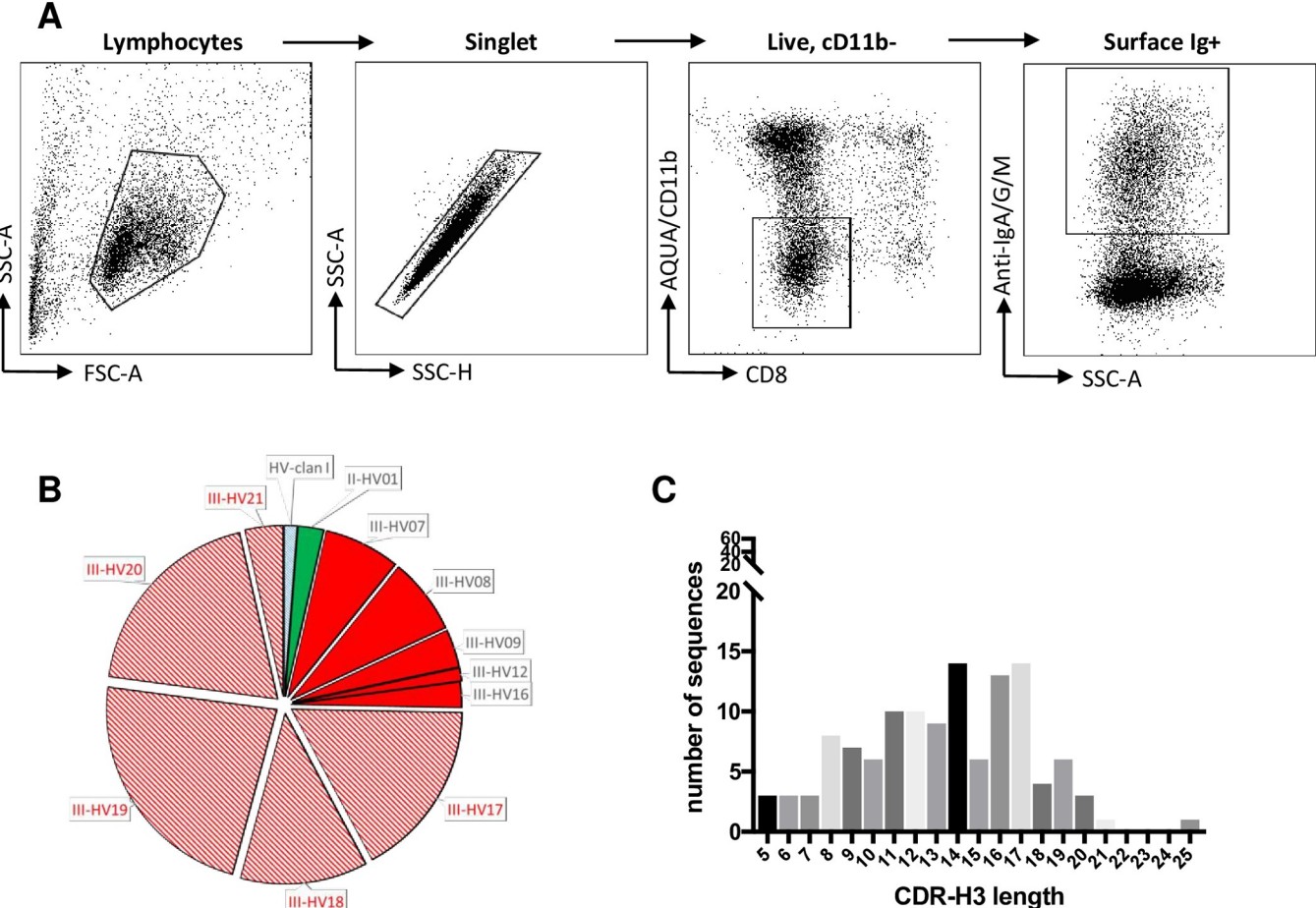

**Fig 3. Genetic features of recovered ferret heavy chain immunoglobulin sequences.** (**A**) Gating scheme for sorting single ferret Bcells for PCR recovery of recombined immunoglobulin genes. (**B**) Distribution of variable germline genes recovered from productive, recombined heavy chain immunoglobulin is shown as pie charts. The width of each segment is proportional to the number of recovered sequences. Sequences corresponding to predicted germlines are shown in solid (less than 1% variable), while sequences with poor alignment to predicted germlines are hatched. Novel potential germlines are indicated in red. (**C**) Distribution of CDR-H3 lengths among recovered immunoglobulin sequences.

recovered (II-LV22, II-LV23) (S1 Table) as were sequences putatively derived from clan V but with limited homology to the annotated genes.

The lengths of the CDR-L3 ranged from 5 to 11 for the kappa locus and 8–13 for the lambda (Fig 4B), both comparable to that of humans and other mammals. Overall, this initial pilot allowed about 12.5% (60/480) recovery efficiency of functional heavy and light chain pairs from single sorted ferret B cells, with 31/60 pairs utilizing Kappa and 29/60 pairs utilizing Lambda chains. Further improvements in the flow cytometric panel for ferret B-cells and annotation of immunoglobulin genes will enhance the recovery of antibody sequences in the future.

## Sequence validation of ferret constant gene segments

Using a next-generation sequencing approach analogous to previous reports [27], we recovered cDNA sequences of ferret constant regions and investigated potential immunoglobulin subclasses. RNA was extracted from ferret splenocytes and subject to RNA-Seq. Putative mRNA transcripts were assembled *de novo* and 5 heavy chain isotypes (IgM, IgG, IgE, IgD and IgA) and two light chain (IgK and IgL) constant genes identified (sequences in S1 Table).

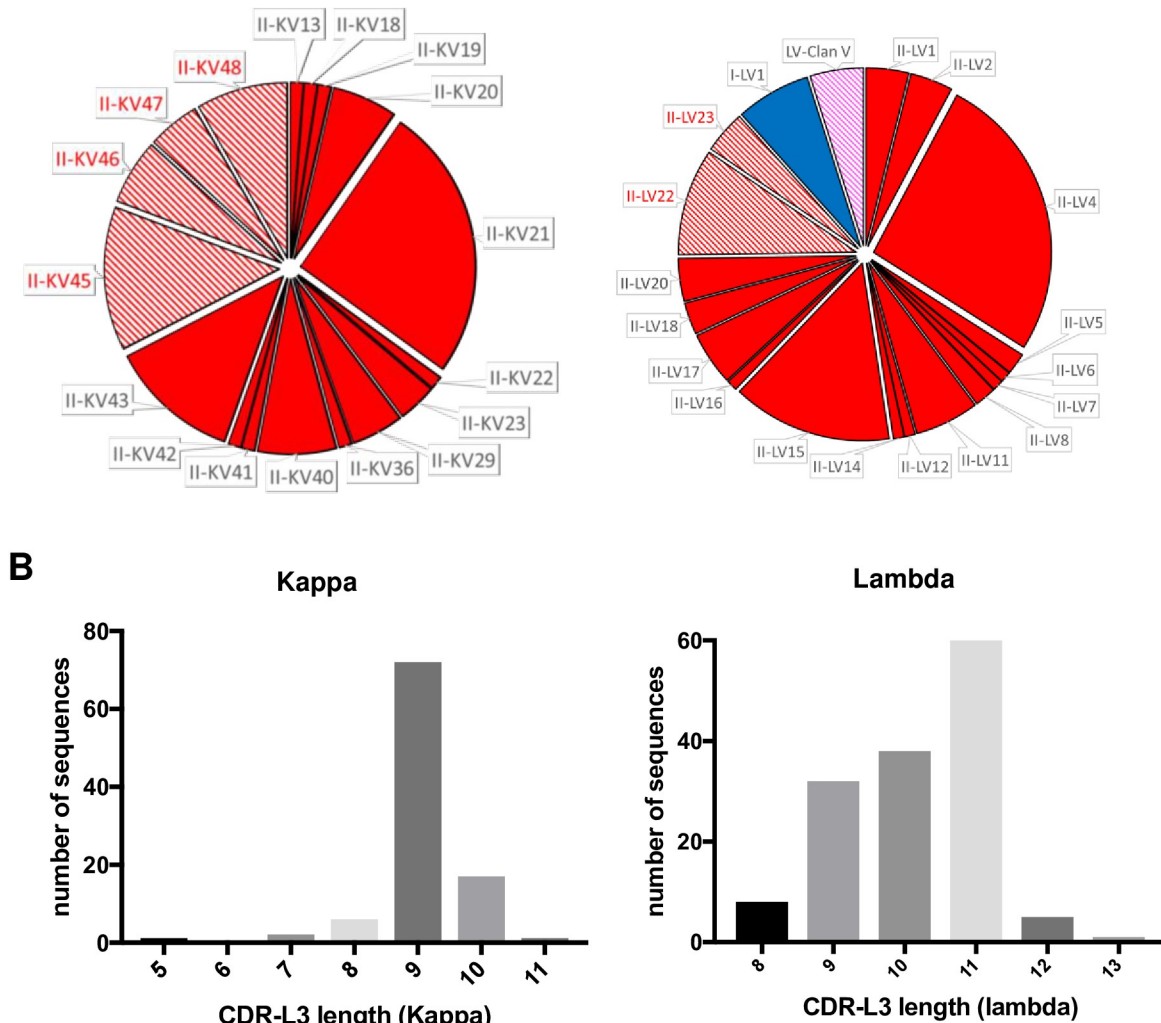

**Fig 4. Genetic features of recovered ferret light chain immunoglobulin sequences.** (**A**) Distribution of variable germline genes recovered from productive, recombined kappa and lambda light chain immunoglobulins are shown as pie charts. The width of each segment is proportional to the number of recovered sequences. Sequences corresponding to predicted germlines are shown in solid (less than 1% variable), while sequences with poor alignment to predicted germlines are hatched. Novel potential germlines are indicated in red. (**B**) Distribution of CDR-L3 lengths among recovered immunoglobulin sequences.

Overall homology to both human and canine sequences was high, with the exception of IgD where the CH1 and hinge domain shows high sequence divergence, consistent with past reports [28]. Minimal variation was observed to artificially spliced genomic sequences, with the exception of some single amino-acid substitutions which could indicate allelic variation within outbred ferrets. Notably, we were unable to identify any IgG subclass variants using this approach, which with four distinct subclasses identified in other carnivores [29], might reflect low transcript abundance within our samples precluding sufficient subclass cDNA recovery.

## Recombinant expression of a chimeric human/ferret monoclonal antibody

We next developed the capacity to recombinantly express and purify ferret IgG. Constant genes for ferret IgG, kappa and lambda chains were synthesised and cloned into mammalian

expression vectors. Recombined human VDJ (heavy) and VJ (lambda) genes from influenza-specific human antibody CR9114 [19] were joined to ferret constant regions to create chimeric IgG. Transfection of heavy and lambda chain plasmids into a mammalian expression system enabled the purification of chimeric ferret/human IgG using standard protein-A purification (Fig 5A), with the resultant antibody retaining HA-specificity (Fig 5B).

## Recovery of hemagglutinin-specific monoclonal antibodies from an influenza-infected ferret

We next sought to recover monoclonal antibodies from influenza immune ferrets. From a single ferret infected with H1N1 A/California/04/2009, we obtained cells from parapharyngeal lymph nodes 28 days post-infection, stained with a panel of ferret B cell surface markers and sorted single cells that bound to recombinant HA probes [15] labelled in two alternate fluorophores (PE and APC) (Fig 6A). Due to an inability to identify ferret IgG subclasses, we designed additional primers (S2 Table) targeting IGHJ-gene segments for multiplex PCR amplification of ferret immunoglobulin genes. From 960 sorted HA specific B-cells, we recovered 263 productive, recombined heavy chain immunoglobulin sequences including significant clonal expansions (Fig 6B). Representative examples from various antibody lineages were cloned into ferret expression vectors, expressed in mammalian cell culture and screened by ELISA. Two antibodies (3B03 and 4A06), derived from a common lineage, were found to both bind to full length A/California/04/2009 HA by ELISA (Fig 6C). Both antibodies (sequences in Fig 6D) mediated potent hemagglutinin inhibition activity against A/California/04/2009 (data not shown) and similarly when tested using an in vitro neutralization assay, both 3B03 and 4A06 were able to prevent A/California/04/2009 infection of MDCK cells down to an effective concentration of 0.4mg/ml and 0.08mg/ml respectively. In order to better understand the epitope on HA recognised by 3B03 and 4A06, we generated escape viruses by serial passaging of

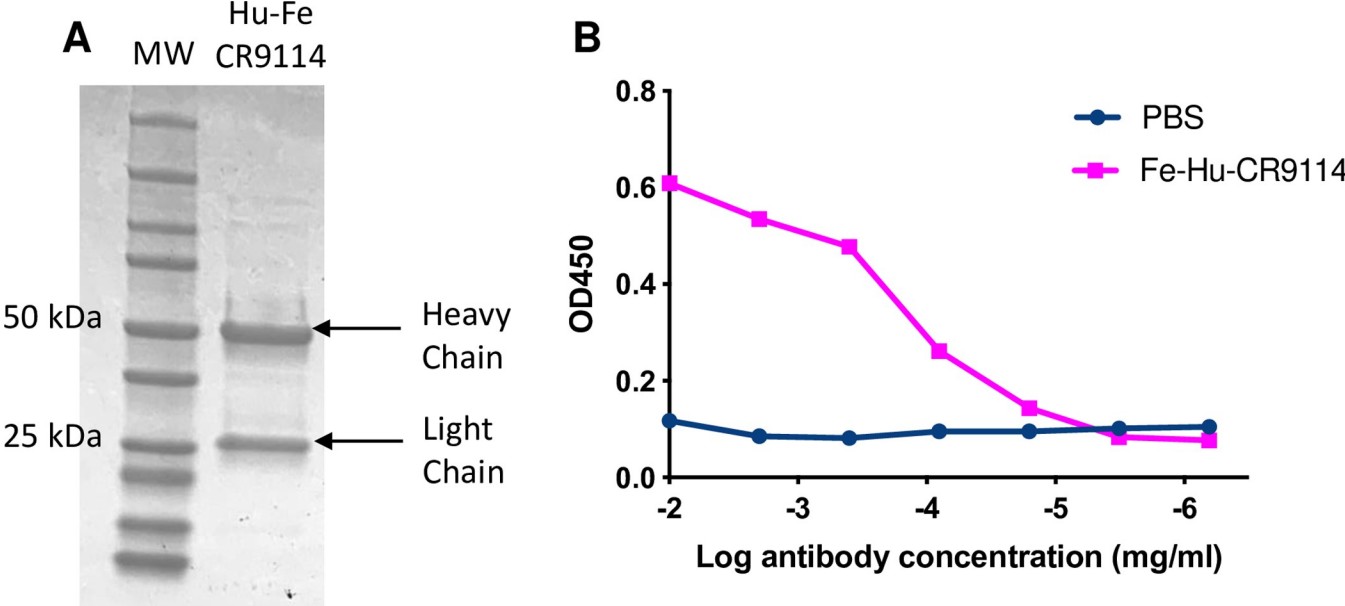

**Fig 5. Recombinant expression of chimeric human/ferret monoclonal antibody (mAb) expressing the CR9114 variable domain.** (A) Reducing SDS-PAGE gel of chimeric CR9114 mAb. Heavy (50kDa) and light chains (25kDa) are indicated. (B) Binding of chimeric CR9114 antibody to full length recombinant A/California/09/2009 HA protein. Ferret-Human CR9114 mAb was serially diluted in PBS to detect A/California/04/2009 HA binding. 1x PBS was included as a negative control (no ab control).

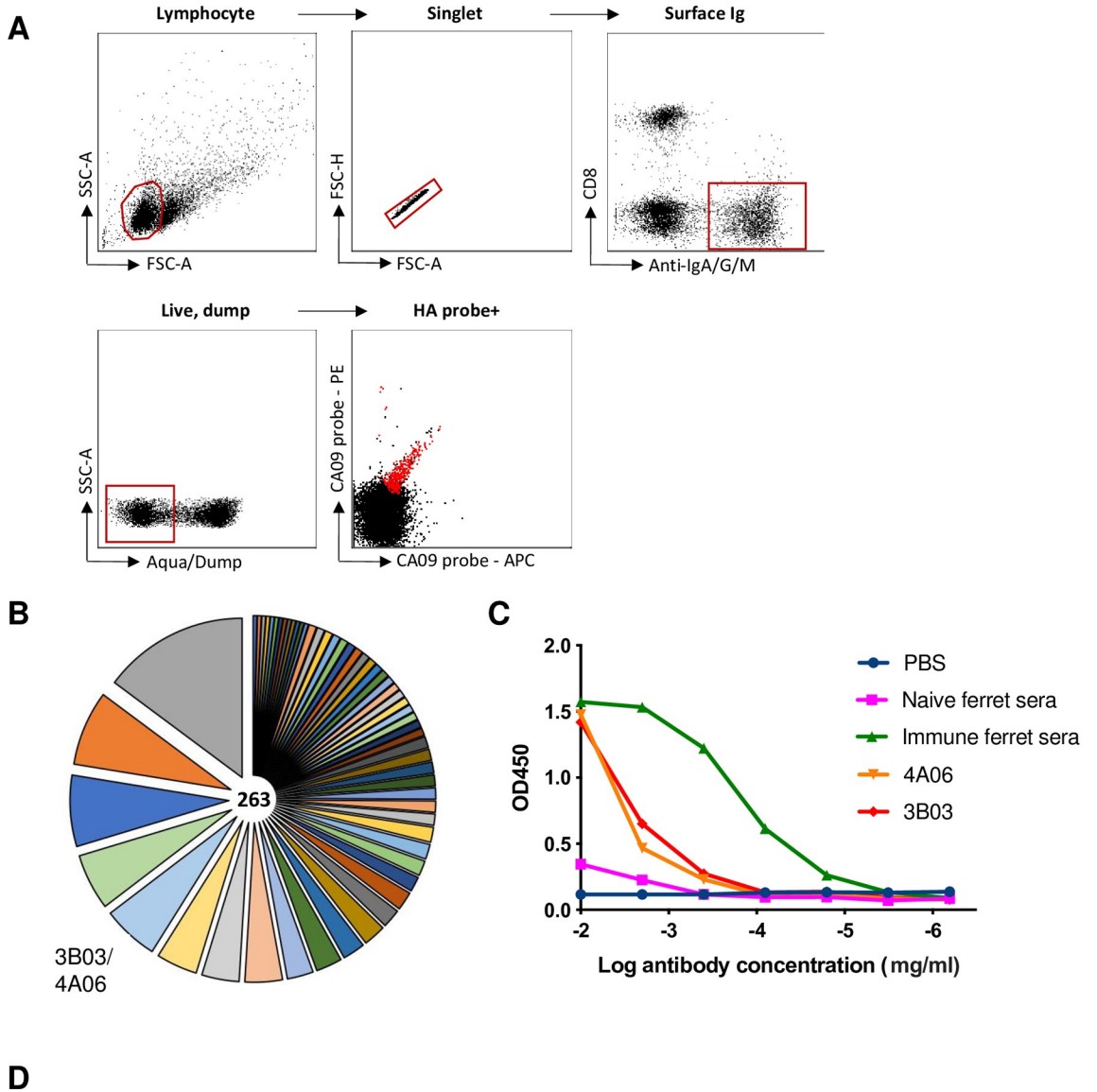

**Fig 6. Recovery and expression of ferret immunoglobulins from HA-specific B cells. (A)** Gating scheme for flow cytometric sorting of single B cells from lymph node suspensions from ferrets infected with A/California/04/2009. Cells binding recombinant HA probes (red) were sorted into 96-well plates for multiplex PCR amplification of heavy and light chain immunoglobulin sequences. (**B**) Clonal distribution of recovered productive, recombined heavy chain immunoglobulins is shown as a pie chart, with each segment representing a distinct clonal family, the width of each segment proportional to the number of recovered sequences and the total number of sequences recovered is indicated. (**C**) Binding of fully-ferret monoclonal antibodies to A/California/09/2009 HA protein was measured by ELISA. Ferret monoclonal antibodies 4A06 and 3B03 or serum samples from immunologically naïve ferrets (naïve serum) or ferrets infected with 1000 TCID$_{50}$ A/California/04/2009 (immune serum) (28 d.p.i) were serially diluted in PBS to detect A/California/04/2009 HA binding. 1x PBS was included as a negative control (no ab control). (**D**) Recombined heavy (VDJ) and lambda (VJ) chain immunoglobulin sequences from recovered HA-specific ferret mAbs. Inferred somatic mutations from germline indicated in red.

A/California/04/2009 in the presence of ferret mAbs followed by sequencing of the full-length HA gene using standard techniques [20]. Neutralisation resistant virus displayed a single K163E mutation (Fig 7) localised within the canonical Sa epitope [30] and previously associated in human populations with escape from pandemic H1N1 serum neutralising activity [31, 32].

## Discussion

We sought to improve the utility of ferrets as an immunological model by deriving techniques for the analysis of B cell immunity. Annotation of genomic immunoglobulin gene segments enabled the design of a multiplex PCR approach to recover ferret BCR sequences. A broad cross-section of predicted germline segments from both heavy and light chain loci were amplified, including several potentially novel germline genes currently absent from available genomic contigs. We noted highly biased V gene utilisation for both heavy and light chain naïve repertoires, with a majority of recovered sequences derived from the most numerous V gene clans (which we termed HV-III, KV-II and LV-II respectively). This observation mirrors the gene distribution in other carnivores, where majority of recovered heavy chain sequences are similarly biased [22, 23, 33, 34]. Light chain bias has been reported for mice (kappa) [35] and both dogs and cats (lambda) [36]. However, humans display more balanced usage of both the kappa and lambda chains [34, 37], as did ferrets in the current study. The distribution of ferret CDR-H3 (mode 13–14 AA) and lambda and kappa CDR-L3 lengths (mode 11 AA and 8–9 AA respectively) were broadly similar to reports from cats [23], dogs [33] and humans [38].

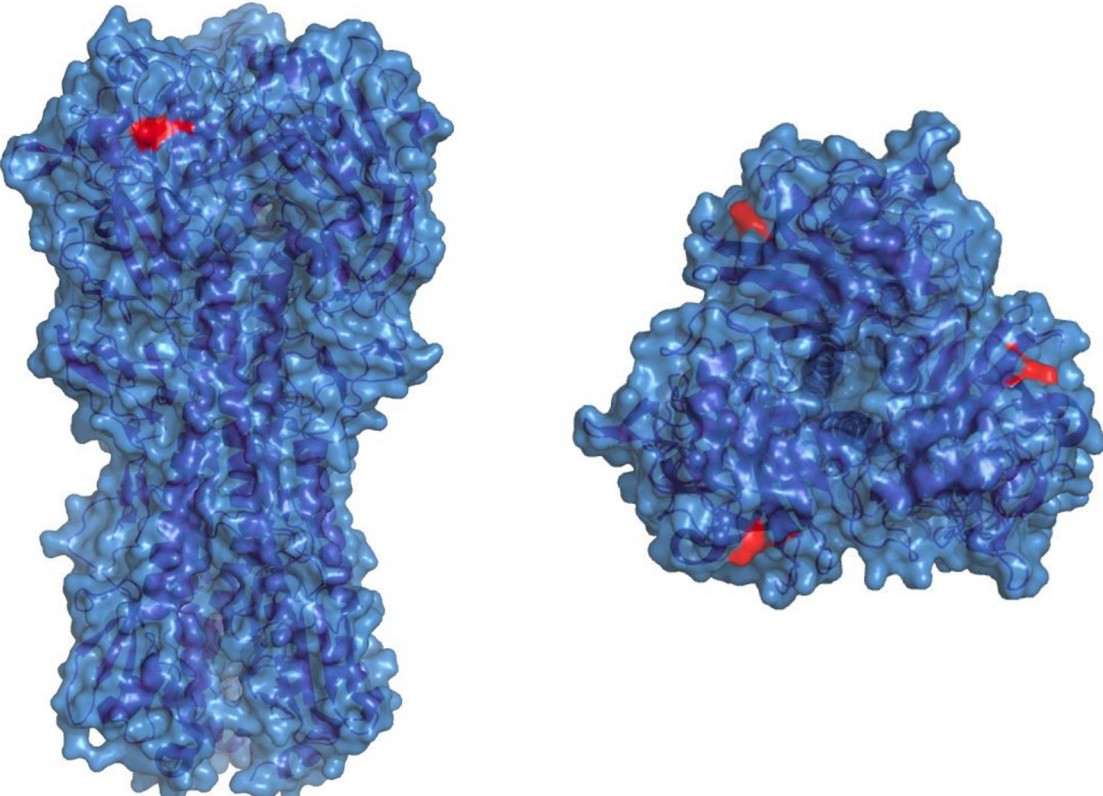

**Fig 7. K163E escape mutation elicited by ferret mAb 4A06 mapped onto A/California/04/20009 (PDB:3LZG).**

Using the new information on ferret Ig sequences, we were able to generate a chimeric ferret/human mAb that retained specificity for influenza HA, and also two novel fully ferret mAbs that exhibited potent binding and neutralisation against A/California/04/2009. These reagents and similar mAbs could be used in repeated immunotherapy studies in ferrets with less concern about the generation of ferret-anti-human responses to human mAbs. We note that while we were successful in the recombinant expression of chimeric human/ferret or fully ferret lambda chain utilising antibodies, further work is needed to generate analogous kappa chain utilising mAbs.

The utility of BCR repertoire analysis to study antigen specific B-cell responses in ferrets will increase as key knowledge gaps are bridged. Firstly, novel ferret-specific flow cytometry reagents are required to enable the resolution of different ferret B-cell populations, in particular, validated pan-B cell lineage surface markers (such as CD19 or CD20) and markers such CD27 and IgD to accurately distinguish memory versus naïve B cell populations. Further development of reagents to identify ferret memory B cells will enhance the resolution of antigen-specific B cell staining and improve recovery of HA-specific ferret mAbs.

Secondly, increased genomic information, particularly, the confirmation of ferret immunoglobulin germline genes and allelic variation among outbred ferrets will allow the development of comprehensive gene databases such as those maintained by International ImMunogeneTics (IMGT) [14]. While we employed a multiplex PCR approach to amplify immunoglobulin genes, the use of template switching based methods such as 5'RACE has been applied to other species such as dogs [39] and may reduce the potential for primer bias driving preferentially recovery of specific immunoglobulin gene segzments. Alternatively, next generation high throughput approaches and tools for the analysis of large RNA-seq data sets such as VDJPuzzle [40], BraCer [41], BALDR [42] and BASIC [43] have been deployed for analysis of antibody repertoires from cats [44] and rhesus macaques [45]. While the entire ferret kappa loci is assembled in the current copy of the ferret genome, the lambda and heavy chain loci were distributed across a number of contigs and the gene arrangement of these two loci remains unclear. Recent advances in long-read NGS approaches such as Oxford Nanopore [46] technologies could enable high resolution mapping of these loci in ferrets, as recently demonstrated by the assembly of reference gene loci in rhesus macaques [45] which facilitates the in depth analysis of antigen-specific immunoglobulin gene repertoires and the characterisation of antigenic epitopes [47]. Additional germline inference methods such as IgDiscover [48] and TIgER [49] could also be deployed in ferrets to more accurately assess and categorise allelic variation within these outbred animals.

Further work clarifying the range of ferret IgG subtypes and the engagement with cellular Fc-receptors (FcR) is required. While we failed to identify IgG subclasses in the current study, the presence of three or four different isotypes in closely related species such as minks [50], cats [51] and dogs [29] suggests these may still exist in ferrets. Recent studies have proposed a critical role for antibody effector functions for protection against viral pathogens such as influenza [52, 53]. As such, further characterisation of ferret IgG, FcR and capacity to mediate ADCC and other Fc-dependent antibody responses is needed.

The HA reactivity of immune ferret sera is a critically important issue since it guides the selection of human influenza strains for inclusion in seasonal influenza vaccines each year [54]. We identified two ferret mAbs, derived from a single clonally expanded family, which displayed anti-HA reactivity and mediated virus neutralisation in vitro. Interestingly, a single mutation (K163E) was able to mediate complete escape from neutralising activity, and similar mutations have been described in both circulating viruses in human populations [55] and from viral escape mutants selected under pressure from human serum [31, 32] or mAbs [56]. Futher cataloguing differential recognition of HA at the mAb level would complement

serological studies, since serum samples from humans, guinea pigs, mice and ferrets have been shown to exhibit differential immunodominance hierarchies within polyclonal responses targeting the canonical epitopes of HA [57] and drive different patterns of viral escape [58]. The tools reported in the current study may also be informative for other emerging human respiratory viruses, most notably the current SARS-CoV2 pandemic. Ferrets can be productively infected with SARS-CoV2 [59, 60] and will serve as a critical model for testing therapeutic options and vaccines against the virus. The capacity to characterise immune repertoires and recover ferret mAbs from SARS-CoV2 infected or immunised ferrets might further support global efforts to develop effective countermeasures against COVID-19.

In summary, we report a methodology to sequence antigen specific B-cells in ferrets, allowing expression of chimeric ferret/human IgG monoclonal antibodies. Further in-depth studies of ferret B-cell repertoires will significantly advance the utility of ferrets as immunological models for critical human diseases such as influenza and SARS-CoV2.

## Supporting information

**S1 Table.**
(XLSX)

**S2 Table.**
(XLSX)

**S1 Raw images. Uncropped gel image.**
(PDF)

## Acknowledgments

The Melbourne WHO Collaborating Centre for Reference and Research on Influenza is supported by the Australian Government Department of Health. JW is supported by a Melbourne International Research Scholarship and Melbourne International Fee Remission Scholarship.

## Author Contributions

**Conceptualization:** Stephen J. Kent, Adam K. Wheatley.

**Data curation:** Julius Wong.

**Funding acquisition:** Aeron C. Hurt, Stephen J. Kent, Adam K. Wheatley.

**Investigation:** Julius Wong, Celeste M. Tai, Hyon-Xhi Tan.

**Methodology:** Celeste M. Tai.

**Resources:** Aeron C. Hurt.

**Supervision:** Aeron C. Hurt, Hyon-Xhi Tan, Stephen J. Kent, Adam K. Wheatley.

**Writing – original draft:** Julius Wong.

**Writing – review & editing:** Julius Wong, Aeron C. Hurt, Hyon-Xhi Tan, Stephen J. Kent, Adam K. Wheatley.

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
