## [Decision Letter · Decision Letter 0]

6 Apr 2020

PONE-D-20-05643

Sequencing B cell receptors from ferrets (Mustela putorius furo)

PLOS ONE

Dear Dr Wheatley,

Thank you for submitting your manuscript to PLOS ONE. After careful consideration, we feel that it has merit but does not fully meet PLOS ONE’s publication criteria as it currently stands. Therefore, we invite you to submit a revised version of the manuscript that addresses the points raised during the review process.

Please consider the points raised by reviewers one and two. While minor, they are important to address. Please provide details on the frequencies of immunoglobulin chains and discuss the coverage as noted by reviewer one. Similarly Reviewer two asks for additional information on the sequences as a figure or table. Also, the constant gene segments are missing from Sup Table 1. And the recommended discussion of other approaches is important to include. A brief discussion on applications to the current pandemic would be interesting to consider. Perhaps utilize this opportunity to better justify figure 5 as reviewer one notes it is somewhat tangential. Finally, while reviewer one recommends publication, please keep in mind that the reviewers do not make decisions on acceptance of manuscripts.

We would appreciate receiving your revised manuscript by May 21 2020 11:59PM. To enhance the reproducibility of your results, we recommend that if applicable you deposit your laboratory protocols in protocols.io, where a protocol can be assigned its own identifier (DOI) such that it can be cited independently in the future. For instructions see: http://journals.plos.org/plosone/s/submission-guidelines#loc-laboratory-protocols

We look forward to receiving your revised manuscript.

Kind regards,

Stephen Mark Tompkins

Academic Editor

PLOS ONE

Journal Requirements:

2. To comply with PLOS ONE submissions requirements, please provide methods of sacrifice in the Methods section of your manuscript.

Reviewers' comments:

Reviewer's Responses to Questions

**Comments to the Author**

1. Is the manuscript technically sound, and do the data support the conclusions?

Reviewer #1: Yes

Reviewer #2: Yes

2. Has the statistical analysis been performed appropriately and rigorously? 

Reviewer #1: N/A

Reviewer #2: Yes

3. Have the authors made all data underlying the findings in their manuscript fully available?

Reviewer #1: Yes

Reviewer #2: No

4. Is the manuscript presented in an intelligible fashion and written in standard English?

Reviewer #1: Yes

Reviewer #2: Yes

5. Review Comments to the Author

Reviewer #1: The manuscript “Sequencing B cell receptors from ferrets (Mustela putorius furo)” describes the development of a multiplex PCR approach to single cell-sequence ferret IgG antibodies. This study builds the framework for more in-depth immunological studies in an extremely important animal model for respiratory infectious diseases. The manuscript is scientifically soloid, well planned, written and relatively easy to follow by anyone in the field. The authors efforts to assemble the ferret immunoglobulin germline repertoire has been long needed, particularly in the Influenza research world, and will certainly pioneer similar studies in other respiratory infectious diseases.

Overall this manuscript is of extreme value for the scientific community and I strongly recommend its publication in PLOSone. Nonetheless I do have some small concerns:

1. The authors report the recovery of IgGH, IgGV� and IgGV� as 121/480, 99/480 and 144/384. Furthermore, it is then stated that 60/480 heavy/light chain pairs were recovered, but it would be important to know if these are equally distributed between � and � light chains. Plate based single cell sorting for BCR amplification is known to be considerably inefficient mostly due to low RNA input or inefficient diversity coverage during amplification. The ratio of successful heavy and light chain pairs to the number of individual heavy or light chain sequences recovered might be a good indicator good diversity coverage.

2. In figure 5 the authors report the expression and purification of a Ferret/human chimeric antibody. In spite of the technological ingenuity of such experiment, it would be much more interesting to actually express and purify a full ferret monoclonal antibody. If using influenza infected animals, it would be easy to prove the specificity of this antibody by ELISA. Standing alone this experiment adds very little to the manuscript and actually distracts the reader from its main message.

Reviewer #2: The manuscript describes the development of multiplex PCR for obtaining ferret Ig genes and generating recombinant monoclonal antibodies. This is an extremely important development to gain a better understanding of immune responses in ferrets, which have been a valuable model for the study of human respiratory infections. The manuscript is clearly written with well-designed experiments.

Some minor comments are:

Using bulk RNA-Seq to obtain constant region genes with multiple sub-classes can be problematic. It is mentioned that the sequences from RNA-Seq showed minimal variation compared to the genomic sequences. It will be helpful if this information is made available in some Figure or Table.

Sequences of Constant gene segments are not provided in the Supplementary Table 1

Line 117 – Should it be supplementary table 2 instead of 1?

The sequence data should be deposited in Genbank

This is a very good start but significant more work is required to obtain a complete repertoire of germline Ig genes. Some discussion on how previous studies using template-switching methods such as BASIC, BALDR, BraCeR and VDJPuzle2 may be implemented for ferrets will be useful. In addition, long read sequencing has helped significantly improve our understanding of Ig loci in rhesus macaques (Ramesh et al, Front. Immunol. 2017 and Cirelli et al, Cell 2019). Discussion on using long-read sequencing and germline inference methods such as IgDiscover, partis and Tigger will also be helpful with respect to future directions in this field. Lastly, this work is extremely relevant with respect to the current pandemic. Discussing how ferrets maybe useful for studying viral infections such as COVID-19 will highlight the importance of this work.

6. PLOS authors have the option to publish the peer review history of their article (what does this mean?). If published, this will include your full peer review and any attached files.

Reviewer #1: Yes: Rodrigo Abreu

Reviewer #2: Yes: Steven Bosinger & Amity Upadhyay

---

## [Author Response · Author response to Decision Letter 0]

28 Apr 2020

Response to Reviewers

We thank the reviewers for their positive comments and suggestions which have significantly strengthened the manuscript. We address each reviewers’ comments below in point-by-point form. 

Reviewer #1: The manuscript “Sequencing B cell receptors from ferrets (Mustela putorius furo)” describes the development of a multiplex PCR approach to single cell-sequence ferret IgG antibodies. This study builds the framework for more in-depth immunological studies in an extremely important animal model for respiratory infectious diseases. The manuscript is scientifically soloid, well planned, written and relatively easy to follow by anyone in the field. The authors efforts to assemble the ferret immunoglobulin germline repertoire has been long needed, particularly in the Influenza research world, and will certainly pioneer similar studies in other respiratory infectious diseases.

Overall this manuscript is of extreme value for the scientific community and I strongly recommend its publication in PLOSone. Nonetheless I do have some small concerns:

1. The authors report the recovery of IgGH, IgGV� and IgGV� as 121/480, 99/480 and 144/384. Furthermore, it is then stated that 60/480 heavy/light chain pairs were recovered, but it would be important to know if these are equally distributed between � and � light chains. Plate based single cell sorting for BCR amplification is known to be considerably inefficient mostly due to low RNA input or inefficient diversity coverage during amplification. The ratio of successful heavy and light chain pairs to the number of individual heavy or light chain sequences recovered might be a good indicator good diversity coverage.

We have updated the manuscript (Lines 296-297) to reflect the relative proportions of kappa versus lambda light chain pairs recovered. This now reads:

“Overall, this initial pilot allowed about 12.5% (60/480) recovery efficiency of functional heavy and light chain pairs from single sorted ferret B cells, with 31/60 pairs utilizing Kappa and 29/60 pairs utilizing Lambda chains.”

2. In figure 5 the authors report the expression and purification of a Ferret/human chimeric antibody. In spite of the technological ingenuity of such experiment, it would be much more interesting to actually express and purify a full ferret monoclonal antibody. If using influenza infected animals, it would be easy to prove the specificity of this antibody by ELISA. Standing alone this experiment adds very little to the manuscript and actually distracts the reader from its main message.

We agree and have undertaken the recovery of a fully-ferret monoclonal antibody and included both ELISA and viral escape data in the revised manuscript. We have updated the Methods with the following:

“For the recovery of antigen-specific ferret B cells, a single ferret was infected with 1000 TCID50 of H1N1 A/California/04/2009 and a single cell suspension of parapharyngeal lymph node cells (pLN) was prepared at 28 days post-infection and cryopreserved in heat-inactivated FCS containing 10% DMSO. Cells were subsequently thawed and stained with Live/Dead Fixable Aqua (Thermo Fisher), surface stains anti-CD11b-BV510 (Biolegend: clone M1/70), anti-ferret IgA/IgM/IgG-FITC (Rockland Immunochemicals cat.618-102-130), anti-CD8 eFluor450 (eBioscience Clone OKT8), anti-ferret CD4 (Layton et al., 2017) conjugated to APC-Cy-7. Biotinylated recombinant full length A/California/04/2009 hemagglutinin (HA) probes (Whittle et al., 2014) conjugated to streptavidin-PE or streptavidin-APC (Invitrogen) were used to sort single HA-specific B cells into 96-well plates and stored at -20oC.”

And:

“Viral Escape Asssay

The generation of viral escape mutants was based upon previously described protocols (Leon et al. 2017). Briefly, 24-well plates were seeded with 2.5 x 105 Madin Darby Canine Kidney (MDCK) cells per well to form confluent monolayers. The next day, serial dilutions of recombinant ferret mAbs were incubated with A/California/04/20090 virus for one hour at 37oC in Flu-media (Dulbecco's Modified Eagle's Medium (DMEM) with 0.8% bovine serum albumin (BSA), 1% penicillin/streptomycin and 0.1% L-1-Tosylamide-2-phenylethyl chloromethyl ketone (TPCK)-treated trypsin), before adding the virus-antibody mixture to MDCK cells. After 2 to 3 days culture media supernatants from wells with visible cytopathic effect were collected and used to infect a fresh monolayer of MDCK cells in the presence of increasing concentrations of mAb. After serial passaging, culture supernatants were harvested, viral RNA was extracted and the HA gene sequenced. Putative mutant viruses were identified based upon sequence comparison to similarly passaged media-only controls.“

We have updated the Results with the following:

“Recovery of hemagglutinin-specific monoclonal antibodies from an influenza-infected ferret

We next sought to recover monoclonal antibodies from influenza immune ferrets. From a single ferret infected with H1N1 A/California/04/2009, we cells from parapharyngeal lymph nodes 28 days post-infection, stained with a panel of ferret B cell surface markers and sorted single cells that bound to recombinant HA probes (Whittle et al., 2014) labelled in two alternate fluorophores (PE and APC) (Figure 6A). Due to an inability to identify ferret IgG subclasses, we designed additional primers (Supplementary table 2) targeting IGHJ-gene segments for multiplex PCR amplification of ferret immunoglobulin genes. From 376 sorted HA specific B-cells, we recovered 263 productive, recombined heavy chain immunoglobulin sequences including significant clonal expansions (Figure 6B). Representative examples from various antibody lineages were cloned into ferret expression vectors, expressed in mammalian cell culture and screened by ELISA. Two antibodies (3B03 and 4A06), derived from a common lineage, were found to both bind to full length A/California/04/2009 HA by ELISA (Figure 6C). Both antibodies (sequences in Figure 6D) mediated potent hemagglutinin inhibition activity against A/California/04/2009 (data not shown) and similarly when tested using an in vitro neutralization assay, both 3B03 and 4A06 were able to prevent A/California/04/2009 infection of MDCK cells down to an effective concentration of 0.4mg/ml and 0.08mg/ml respectively. In order to better understand the epitope on HA recognised by 3B03 and 4A06, we generated escape viruses by serial passaging of A/California/04/2009 in the presence of ferret mAbs followed by sequencing of the full-length HA gene using standard techniques (Leon et al 2017). Neutralisation resistant virus displayed a single K163E mutation (Figure 7) localised within the canonical Sa epitope (Caton et al. 1982) and previously associated in human populations with escape from pandemic H1N1 serum neutralising activity (Chengjun et al. 2016; Davis et al. 2018).”

And we have included a new Figures 6 and 7:

And additional text has been incorporated throughout the discussion.

Reviewer #2: The manuscript describes the development of multiplex PCR for obtaining ferret Ig genes and generating recombinant monoclonal antibodies. This is an extremely important development to gain a better understanding of immune responses in ferrets, which have been a valuable model for the study of human respiratory infections. The manuscript is clearly written with well-designed experiments.

Some minor comments are:

Using bulk RNA-Seq to obtain constant region genes with multiple sub-classes can be problematic. It is mentioned that the sequences from RNA-Seq showed minimal variation compared to the genomic sequences. It will be helpful if this information is made available in some Figure or Table.

Sequences of Constant gene segments are not provided in the Supplementary Table 1

Line 117 – Should it be supplementary table 2 instead of 1?

Sequences for the ferret constant gene segments have been included within Supplementary Table 1 (Tab – “constant”)

The sequence data should be deposited in Genbank

Relevant sequence data have been deposited to Genbank (submission ID:2331742)

This is a very good start but significant more work is required to obtain a complete repertoire of germline Ig genes. Some discussion on how previous studies using template-switching methods such as BASIC, BALDR, BraCeR and VDJPuzle2 may be implemented for ferrets will be useful. In addition, long read sequencing has helped significantly improve our understanding of Ig loci in rhesus macaques (Ramesh et al, Front. Immunol. 2017 and Cirelli et al, Cell 2019). Discussion on using long-read sequencing and germline inference methods such as IgDiscover, partis and Tigger will also be helpful with respect to future directions in this field. 

We have expanded the discussion to cover these important sequencing technologies. It now includes the following section:

“While we employed a multiplex PCR approach to amplify immunoglobulin genes, the use of template switching based methods such as 5’RACE has been applied to other species such as dogs (39) and may reduce the potential for primer bias driving preferentially recovery of specific immunoglobulin gene segzments. Alternatively, next generation high throughput approaches and tools for the analysis of large RNA-seq data sets such as VDJPuzzle(40), BraCer(41), BALDR(42) and BASIC(43) have been deployed for analysis of antibody repertoires from cats (44) and rhesus macaques (45). While the entire ferret kappa loci is assembled in the current copy of the ferret genome, the lambda and heavy chain loci were distributed across a number of contigs and the gene arrangement of these two loci remains unclear. Recent advances in long-read NGS approaches such as Oxford Nanopore (46) technologies could enable high resolution mapping of these loci in ferrets, as recently demonstrated by the assembly of reference gene loci in rhesus macaques (45) which facilitates the in depth analysis of antigen-specific immunoglobulin gene repertoires and the characterisation of antigenic epitopes (47). Additional germline inference methods such as IgDiscover (48) and TIgER (49) could also be deployed in ferrets to more accurately assess and categorise allelic variation within these outbred animals.”

Lastly, this work is extremely relevant with respect to the current pandemic. Discussing how ferrets maybe useful for studying viral infections such as COVID-19 will highlight the importance of this work.

We have expanded the discussion to cover the importance of ferrets for coronavirus research. It now includes the following section:

“The tools reported in the current study may also be informative for other emerging human respiratory viruses, most notably the current SARS-CoV2 pandemic. Ferrets can be productively infected with SARS-CoV2 (59, 60) and will serve as a critical model for testing therapeutic options and vaccines against the virus. The capacity to characterise immune repertoires and recover ferret mAbs from SARS-CoV2 infected or immunised ferrets might further support global efforts to develop effective countermeasures against COVID-19.”

---

## [Editor Report · Decision Letter 1]

13 May 2020

Sequencing B cell receptors from ferrets (Mustela putorius furo)

PONE-D-20-05643R1

Dear Dr. Wheatley,

We are pleased to inform you that your manuscript has been judged scientifically suitable for publication and will be formally accepted for publication once it complies with all outstanding technical requirements.

With kind regards,

Stephen Mark Tompkins

Academic Editor

PLOS ONE
---

## [Editor Report · Acceptance letter]

18 May 2020

PONE-D-20-05643R1 

Sequencing B cell receptors from ferrets (*Mustela putorius furo*) 

Dear Dr. Wheatley:

I am pleased to inform you that your manuscript has been deemed suitable for publication in PLOS ONE. Congratulations! Your manuscript is now with our production department. 

With kind regards,

on behalf of

Dr. Stephen Mark Tompkins 

Academic Editor

PLOS ONE